# Cognitive Stimulation and Its Effects on Well-Being, Executive Functions, and Brain-Derived Neurotrophic Factor in Older Adults from a Mexican Geriatric Center: A Quasi-Experimental Study

**DOI:** 10.3390/nursrep15050151

**Published:** 2025-04-30

**Authors:** Nadia Yanet Cortés-Álvarez, César Rubén Vuelvas-Olmos, Leticia Gabriela Marmolejo-Murillo, Elizabeth Sánchez-Duarte, Alfredo Lara-Morales

**Affiliations:** 1Department of Nursing and Midwifery, Division of Natural and Exact Sciences, University of Guanajuato, León 37150, Mexico; ny.cortes@ugto.mx; 2Medical Sciences Program, School of Medicine, University of Colima, Colima 28040, Mexico; czarvuelvas@gmail.com; 3Department of Medicine and Nutrition, Division of Health Sciences, University of Guanajuato, León 37150, Mexico; lg.marmolejo@ugto.mx; 4Department of Applied Sciences to Labor, Division of Health Sciences, University of Guanajuato, León 37150, Mexico; elizabeth.sanchez@ugto.mx

**Keywords:** cognitive stimulation, executive functions, well-being, Brain-Derived Neurotrophic Factor, older adults, geriatric center, nursing

## Abstract

**Background/Objectives:** The progressive increase in the aging population highlights the need for interventions aimed at preserving cognitive health and overall well-being in older adults. This study aimed to assess the impact of a structured cognitive training program on psychological well-being, executive function performance, and Brain-Derived Neurotrophic Factor levels (BDNF) in older adults from a Mexican geriatric center. **Methods:** A quasi-experimental pretest–posttest design with a nonequivalent control group was conducted. Thirty-two older adults were assigned either to a cognitive stimulation intervention group or a control group. The intervention consisted of 120 individually structured sessions, each lasting approximately 60 min, delivered five times per week over 24 weeks. Independent neuropsychologists, blinded to group allocation, assessed executive function (BANFE-3), depressive symptoms (Yesavage Geriatric Depression Scale), autonomy in daily living (Barthel Index), and quality of life (WHOQOL-OLD) before and after the intervention. Serum BDNF levels were also measured. **Results:** The intervention group showed significant improvements in executive function, depressive symptoms, independence in daily activities, and quality of life, while the control group showed no changes. Additionally, the intervention group showed an increase in BDNF expression post-intervention. **Conclusions:** The cognitive stimulation program effectively improved cognitive performance, emotional well-being, autonomy, and quality of life in older adults. These findings highlight the importance of integrating structured cognitive stimulation into geriatric care. For nursing practice, this underscores the key role nurses can play in delivering cognitive interventions to promote cognitive health, independence, and emotional stability among institutionalized and non-institutionalized older adults.

## 1. Introduction

Population aging, resulting from increased life expectancy and declining birth rates, is a global phenomenon that presents significant challenges for health systems and public policy. Worldwide, the proportion of individuals over the age of sixty is rising rapidly, with estimates indicating that by 2030, one in six people will belong to this age group [1]. This demographic shift poses substantial public health challenges, particularly due to the cognitive changes associated with aging [2]. One of the most affected cognitive domains in this process is executive functions, a specialized subset of cognitive functions primarily associated with the brain’s prefrontal cortex, which are crucial for goal-directed behavior and self-regulation [3].

Cognitive decline, particularly in executive functions, can significantly impact the well-being of older adults [4]. The decline of these abilities can significantly impair the capacity to perform basic activities of daily living, thereby reducing functional autonomy [5]. This loss of autonomy is closely associated with an increased risk of developing depressive symptoms, as functional dependence and perceived incapacity negatively impact emotional health [6]. Collectively, cognitive decline, reduced autonomy, and the presence of depression contribute to a marked decrease in quality of life among older adults [7]. These relationships highlight the importance of developing interventions aimed at preserving and enhancing cognitive functions to promote healthy aging and sustained well-being.

Cognitive stimulation (CS) has emerged as a promising non-pharmacological intervention for preserving and enhancing cognitive functions in older adults. It involves a structured set of activities and exercises specifically designed to promote, maintain, and improve several cognitive domains, including memory, attention, and executive functions [8,9]. However, studies analyzing the effects of cognitive stimulation on cognitive functions have yielded inconsistent results. While some studies found a significant increase in participants’ cognitive function after the intervention compared to the control group [10,11,12,13,14], others failed to find an improvement in cognitive function [15,16,17], and some even found an increase in cognitive decline [18].

Furthermore, most CS programs are of short duration, highlighting the relevance of prolonged interventions as a key factor in optimizing their effectiveness. Longer interventions enable more comprehensive follow-up of participants and provide sufficient time for the consolidation and stabilization of cognitive improvements in older adults [13,19]. In support, Chiu et al. [20] suggested that the intervention characteristics of ≧3 times each week, ≧8 total training weeks, and ≧24 total training sessions yield a higher effect size. In addition, several considerations could enhance the scope and effectiveness of CS programs: (1) the use of gamification in delivering interventions may increase participant adherence and engagement [21]; (2) implementing the intervention in an individual format could better accommodate each participant’s pace and provide access for those unable to participate in group settings due to personal preferences, health conditions, behavioral issues, or fear of stigmatization [21,22] and (3) analyzing predictors of intervention outcomes and adherence could contribute to the optimization and personalization of future interventions [21].

CS has been shown to play a significant role in promoting neuroplasticity, the brain’s ability to adapt and reorganize in response to new experiences and learning throughout the lifespan. This neuroplastic process is supported by several molecular mechanisms, among which the Brain-Derived Neurotrophic Factor (BDNF) is particularly important [23]. BDNF is a neurotrophin that facilitates neuronal survival, synaptic plasticity, and the formation of new neural connections, all of which are essential for cognitive function and learning [23]. Evidence suggests that engaging in structured cognitive stimulation activities can increase BDNF levels [24], thereby enhancing neuroplasticity and contributing to the maintenance and improvement of cognitive functions in older adults [24].

Given the imminent inversion of the population pyramid and the growing need to address the cognitive needs of this vulnerable population, effective strategies to improve or mitigate cognitive deficiencies in older adults are essential. Therefore, the primary aim of this study was to evaluate the impact of a cognitive training program on older adults, specifically assessing its effects on psychological well-being, executive function performance, and serum BDNF levels. The secondary aim was to explore the potential association between these cognitive changes and variations in BDNF expression, contributing to a better understanding of the neurobiological mechanisms underlying cognitive stimulation and their role in promoting healthy aging.

## 2. Materials and Methods

### 2.1. Design

A quasi-experimental, pretest–posttest design was conducted with a nonequivalent control group, with a 1:2 allocation.

Participants were randomly assigned to one of two study groups: the experimental group (participants who received the intervention) and control group (participants with similar characteristics who did not receive the intervention). Throughout the study period, both groups continued to receive their usual medical and therapeutic care. All assessment instruments were administered simultaneously to both groups to ensure consistency in evaluation conditions.

The study design, data collection, analysis, and reporting were conducted in accordance with the STROBE (Strengthening the Reporting of Observational Studies in Epidemiology) guidelines [25] (see Appendix A). Regarding ethical considerations, this research was conducted in accordance with the Declaration of Helsinki, the General Health Law on Research, and was approved by the Research Ethics Committee of the San Luis de la Paz General Hospital (CONBIOÉTICA Registry: CONBIOÉTICA-11-CEI-003-20200312). Participation was entirely voluntary, without any financial or other incentives.

### 2.2. Sample Size

Based on a previous study [26], a sample size of 36 participants in total was estimated to be sufficient to detect a mean difference of 1.05 between the experimental and control groups, assuming an α of 0.05 and a power (1 − β) of 0.8. Considering a 10% dropout rate, the recruitment target was set at 12 participants per group, maintaining a 2:1 ratio of experimental to control participants.

### 2.3. Participants

Older adults who attended the Gerontological Centers of the National System for the Integral Development of Families (DIF) in Guanajuato, Mexico, were invited to participate. (DIF is a Mexican decentralized agency attached to the Ministry of Health, responsible for directing social assistance at the national level.) Participants were randomly assigned to either the intervention or control group. To ensure comparability, randomization was stratified based on gender, age, educational level, and pre-intervention cognitive function score. This approach aimed to balance key characteristics between groups, minimizing potential confounding factors. To participate in the study, participants had to meet the following selection criteria:

Inclusion:

○Adults (of any gender) over 60 years of age.○Willingness to participate in all intervention and assessment sessions.○Ability to communicate verbally.○Ability to participate in simple games/activities.

Exclusion

○Impairments or behavioral problems that could prevent participation in activities.○Have received psychological or psychiatric care in the last two months.○Presenting any condition that requires immediate intervention (i.e., suicidal ideation) or that interferes with participation in the study (i.e., severe hearing loss).○Inability to communicate that limits participation in the intervention, as determined by the researchers.○Presence of any medical condition that endangered survival during the project.

Elimination

○Change in residence.○Participant or family member’s decision to withdraw from the study.○Participating in less than 80% of the intervention sessions.

### 2.4. Blood Sample Collection and Measurement of BDNF Serum Levels

Venous blood samples were obtained from participants after an overnight fast of at least 8 to 12 h by puncture of the antecubital vein. Collection was performed using yellow tubes (tubes with separating gel and without anticoagulant; BD Vacutainer^®^ SST™, Franklin Lakes, NJ, USA). The samples were allowed to stand at room temperature for 30 min to allow clotting and then centrifuged at 3000 rpm for 10 min. The resulting serum was processed by sandwich enzyme-linked immunosorbent assay (ELISA technique) using a commercial kit (Human BDNF ELISA Kit (ab212166, Abcam, Cambridge, MA, USA)) according to the manufacturer’s instructions.

### 2.5. Cognitive Assessment

After the blood sample was collected, participants were allowed to have breakfast and consume their regular medication. Subsequently, and after a minimum period of one hour had elapsed, the cognitive assessment was conducted. Two neuropsychologists who were unaware of the study aims and the condition assigned to each participant conducted assessments at both pre-intervention (baseline) and post-intervention (6 months). The Neuropsychological Executive Functions and Frontal Lobes battery-3 (in Spanish, Batería Neuropsicológica de Funciones Ejecutivas y Lóbulos Frontales, BANFE-3), a neuropsychological battery that had been designed and validated for the Mexican population, was used [27], ensuring its applicability and accuracy in assessing executive functions in the population studied.

This instrument comprises a comprehensive set of tests with high reliability and validity for cognitive processes evaluation that depend mainly on the prefrontal cortex. It provides a global performance index of the functioning of the three prefrontal areas evaluated: the orbitofrontal cortex, anterior prefrontal cortex, and the dorsolateral cortex; it indicates the abilities and inabilities of participants in each of these cognitive areas. These areas are linked to cognitive flexibility, inhibitory control, working memory, planning and organization, problem-solving, reasoning, attention, concentration, processing speed, and decision-making [28].

The interpretation of the total score and each of the areas enables classification of an individual’s performance as follows: high normal (116 and above), normal (85–115), moderate mild impairment (70–84), or severe impairment (<69), with high reliability and a Cronbach’s alpha greater than 0.80.

### 2.6. Measuring Well-Being Indicators

The day after the cognitive assessment was conducted, well-being indicators were assessed. Similar to the cognitive assessment, the assessments were conducted by two neuropsychologists who were unaware of the study aims and the condition assigned to each participant.

To assess symptoms of depression in the past 15 days, the Yesavage Geriatric Depression Scale (GDS-15) was used. The score ranges from 0 to 15, where a score <5 indicates a normal range, 5 to 9 suggests mild depression, and more than 10 indicates moderate to severe depression [29]. This scale was validated in the Mexican population and is considered reliable and valid for measuring depression in older Mexican adults (Cronbach’s alpha = 0.84) [30].

In addition, to assess autonomy in activities of daily living (ADL), the Barthel Index was used, which has been widely validated and used in the geriatric population. This index analyses 10 aspects: bowel, bladder, personal hygiene, bathroom use, feeding, transferring, mobility, dressing, stairs, and bathing. The total score ranges from 0 to 100 points. The higher the score, the greater the functional independence [31]. This instrument has been validated in Mexican older adults and is recognized for its strong psychometric properties, making it a reliable and valid tool for assessing functional independence in the Mexican geriatric population (Cronbach’s alpha = 0.89) [32,33].

Finally, to assess quality of life, the World Health Organization Quality of Life-Old scale (WHOQOL-OLD) was used, an instrument developed by the World Health Organization (WHO) specifically to measure quality of life in older people. It is an instrument made up of 24 items that are answered on a five-point Likert scale, enabling the assessment of various degrees of satisfaction and perceived quality of life; divided into 6 areas: sensory skills, autonomy, past, present and future activities, social participation, death and intimacy [34]. The WHOQOL-OLD was validated in Mexican older adults, determining it suitable for application in the Mexican population to evaluate the construct of quality of life Cronbach’s alpha = 0.89) [35].

### 2.7. Cognitive Stimulation Intervention

A cognitive stimulation program was designed based on the following principles: (1) Cognitive stimulation therapy principles are person-centered, respect, participation, inclusion, choice, fun, maximizing potential, and strengthening individual social relationships between the therapist and each participant [36]; (2) Cognitive reserve is understood as the brain’s ability to cope with and/or tolerate brain changes associated with normal aging or due to a pathological process, delaying or decreasing the symptoms or clinical manifestations [37]. Cognitive reserve is dynamic and multifactorial, which enables participants to maintain functionality despite age-related brain changes [38]; (3) Neuroplasticity, according to which the brain could change as a result of experience [39]; and (4) this intervention followed the guidelines proposed by Dreer et al. [40] to maximize the success of interventions with elderly patients based on their neuropsychological functioning. This includes a higher frequency of sessions with shorter duration, a clear structure, and adaptations to the pace of each participant and the slower information processing speed often observed in older adults.

The main aim of this intervention was to enhance cognitive domains, primarily targeting cognitive flexibility, inhibitory control, working memory, planning, reasoning, attention, concentration, and processing speed. All sessions followed the basic structure of the individual cognitive stimulation program suggested by Justo-Henriques, 2021 [41], as described in Table 1.

During the time allocated for cognitive stimulation, cognitive domains were trained through two cognitive stimulation tools:

1. Paper activities: A cognitive training manual was developed for the paper-based activities, which included 28 types of activities. Each activity comprised 30 exercises, divided into three predefined levels of complexity (basic, intermediate, and advanced), designed to provide a progressive challenge to cognitive functions: level 1 with 10 basic exercises, level 2 with 10 intermediate exercises, and level 3 with 10 advanced exercises (Table 2). In total, the manual contained 840 exercises (28 activities × 30 exercises). For reference, Appendix A includes an example of a Level 1 exercise for each of the 28 activities.

2. Board games: Several board games were used as a tool for cognitive stimulation in older adults (Table 3). Cognitive flexibility was stimulated through games that promoted adaptation to changes and the generation of new strategies. For inhibitory control, activities were implemented that challenged participants to regulate impulses and maintain focus on established rules. Working memory was exercised through games that facilitated the retention and manipulation of short-term information. Planning was stimulated through activities that required players to anticipate moves and develop strategies. Reasoning was promoted through games that encouraged problem-solving and logical decision-making. Attention was targeted through tasks that required sustained focus and precision in task execution. Finally, concentration and processing speed were enhanced through activities that incentivized quick and efficient responses. These games were integrated to create a dynamic cognitive stimulation program.

The intervention was applied during 120 individual sessions of 60 min duration, with a frequency of five times a week (total duration = 6 months or 24 weeks). Appendix B shows the exercise schedule used during the 120 sessions. The intervention was applied from January to July 2024, by the principal investigators as well as by nursing and midwifery students, previously trained by two experts in clinical psychology with six years of experience in cognitive stimulation, at the Gerontological Center.

### 2.8. Statistical Analysis

Statistical analyses were performed using SPSS v.26. Descriptive statistics were computed to analyze the results of the assessment instruments, including measures of central tendency and dispersion. To evaluate changes in cognitive performance before and after the cognitive stimulation program, paired-sample comparison tests were applied, including Student’s *t*-test for related samples or the Wilcoxon signed-rank test when parametric assumptions were not met. Additionally, a comparative analysis of cognitive assessment results and well-being variables before and after the intervention was performed. To account for multiple comparisons and reduce the risk of Type I error, the Bonferroni correction was applied where appropriate.

To assess baseline homogeneity of categorical variables across conditions, the χ^2^ test was used. When expected values were below 5, Fisher’s exact test or the Fisher–Freeman–Halton exact test was applied. For continuous variables, the Mann–Whitney U test was used to compare two independent samples. The analyses followed the intention-to-treat principle, with all participants analyzed in their originally assigned groups. Missing data related to cognitive performance, depressive symptoms, and autonomy in daily living activities were addressed using the last observation carried forward approach.

To examine inter-group differences in cognitive performance, depressive symptoms, and autonomy at pre- and post-intervention, as well as therapist-related outcome variations, the Mann–Whitney U test for independent samples was performed. Intra-group changes from pre- to post-intervention were assessed using the Wilcoxon signed-rank test. Effect sizes were calculated using Cohen’s d, with interpretation thresholds set at d = 0.2 (small), d = 0.5 (moderate), and d = 0.8 (large).

Adherence to the intervention was evaluated by analyzing the frequency distribution of participant dropouts and comparing dropout rates between groups using the Mann–Whitney U test. Additionally, descriptive statistics and frequency distributions were computed to assess the number of attended sessions.

## 3. Results

As presented in Table 4, the final sample comprised 36 participants, with 24 participants in the intervention group and 12 in the control group. The analysis revealed no statistically significant differences between the groups concerning the sociodemographic variables examined. This suggests that the groups were comparable at baseline, enhancing the validity of subsequent comparisons.

### 3.1. Adherence to the Intervention

Out of the 120 sessions that comprised the intervention, participants in the intervention group attended an average of 114.8 ± 5.2 sessions. A total of 28 participants (87.5%) attended all scheduled sessions, while 4 participants (12.5%) attended more than 80% of the sessions.

### 3.2. Cognitive Assessments

#### 3.2.1. Intragroup Differences

As illustrated in Table 5, a comparison of pre-intervention and post-intervention scores demonstrated statistically significant improvements in the total executive functions score and across all subdomains within the intervention group. Additionally, regarding performance classification, the findings indicated significant differences, with an increase in the number of participants exhibiting higher performance levels. In contrast, no significant differences were found in the control group, underscoring the effectiveness of the cognitive stimulation intervention.

#### 3.2.2. Intergroup Differences

A comparison of post-evaluation data revealed a significant improvement in the total executive function score and all individual domains in the intervention group, compared to the control group. Furthermore, regarding performance classification, significant differences were observed, with an increase in the number of participants attaining higher performance levels within the intervention group (Table 6). These results highlight the effectiveness of the cognitive stimulation program in improving executive function in older adults.

### 3.3. Well-Being Indicators

#### 3.3.1. Depression

In relation to the Yesavage scale, a significant difference was observed between the intervention and control groups (t = 2.45, *p* = 0.027). This finding suggests that the intervention had an effect in reducing depression-related symptoms in the evaluated population (Figure 1).

#### 3.3.2. Autonomy in Activities of Daily Living (ADL)

A significant increase in the Barthel Index score was observed between the intervention and control groups (t = 3.534, *p* < 0.01). These results indicate that the cognitive stimulation program effectively enhances the level of independence in activities of daily living (ADL) within this population (Figure 2).

#### 3.3.3. Quality of Life

Quality of life in older adults was assessed using the WHOQOL-OLD questionnaire. Significant differences were found between the intervention and control group comparisons for sensory abilities (t = 2.823, *p* = 0.014), social participation (t = 2.173, *p* = 0.027), and autonomy (t = 3.758, *p* = 0.007). These findings suggest that the intervention had a positive impact on specific aspects of quality of life in older adults (Figure 3).

### 3.4. BDNF Serum Levels

As shown in Figure 4, the intervention group exhibited a significantly higher BDNF expression (362.75 ± 68.79 pg/mL) compared to the control group of older adults (243.17 ± 84.49 pg/mL; t = 3.248, df = 46, *p* = 0.001).

## 4. Discussion

This study evaluated the efficacy of a cognitive stimulation program in an individual format with continued exposure to cognitive stimulation activities in Mexican older adults. Findings show that long-term cognitive stimulation intervention led to significant improvements in executive function within the intervention group. Intragroup comparisons showed a marked increase in the total executive function score and all individual domains, along with a higher proportion of participants achieving higher performance classifications. In contrast, the control group exhibited no significant changes in any of the measured outcomes. Furthermore, intergroup analysis confirmed that the intervention group outperformed the control group in post-intervention assessments, with statistically significant differences in both the overall executive function score and the classification of participants into higher performance categories. These results indicate the intervention’s effectiveness in enhancing cognitive performance and highlight the strength of these effects, suggesting potential clinical relevance in improving functional outcomes. Distinguishing between statistical significance and the actual magnitude of the changes observed will be important for understanding the real-world impact of these improvements.

Our findings are consistent with the idea that cognitive stimulation programs lead to improvements in several areas of cognitive functioning. For example, a systematic review and meta-analysis by Gómez-Soria et al. [42] suggest that cognitive stimulation can increase general cognitive functioning, memory, orientation, praxis, and calculation in older adults. Similarly, the systematic review and meta-analysis by Yun et al. [43] concluded that cognitive-based interventions were effective in improving the cognitive function of older adults, both in those without cognitive impairment and in those with mild impairment, which supports the results of our study by demonstrating significant improvements in these functions after the intervention in the experimental group [9]. These observed benefits may be explained by cognitive-based interventions stimulating neural plasticity, leading to enhanced cognitive performance. Additionally, such interventions might improve mood and motivation, further contributing to cognitive enhancement.

Several mechanisms can explain the neurobiological basis for these cognitive improvements. Cognitive stimulation has been shown to enhance neuroplasticity, which refers to the brain’s ability to reorganize itself by forming new neural connections [39]. This process is crucial for maintaining and improving cognitive functions in aging. Research indicates that cognitive training can increase the density of gray matter in brain regions associated with executive functions, such as the prefrontal cortex. Moreover, such interventions may boost synaptic density and connectivity, contributing to more efficient cognitive processing. Additionally, cognitive stimulation can help mitigate age-related declines in brain volume and support the preservation of cognitive reserves, which are vital for maintaining cognitive abilities despite the effects of aging [37,38]. These neurobiological changes likely underlie the observed improvements in cognitive performance and executive functions in our study.

In this study, a significant increase in serum BDNF levels was observed in the intervention group compared to the control group. BDNF is a key neurotrophic factor that plays an essential role in synaptic plasticity and neuronal survival, and its increased expression is associated with increased cognitive resilience. These findings are consistent with previous studies that have shown that cognitive training can favor BDNF upregulation, thus promoting cognitive function and contributing to the slowing of neurodegenerative processes [24,44]. For instance, a study by Ledreux et al. demonstrated that cognitive training over a 5-week period was associated with increased serum BDNF levels in older adults, suggesting benefits for brain health with aging [44]. Similarly, recent studies have shown that traditional Chinese exercises may improve BDNF levels in older populations, although variations in study designs necessitate further confirmation. However, differences in BDNF elevation across studies may be attributed to variations in training intensity, duration, and participant characteristics [45,46,47]. While some research indicates that combining physical and cognitive activities produces greater BDNF increases, our study demonstrates that cognitive training alone can have a significant neurotrophic effect. Additionally, methods of BDNF assessment and timing of sample collection may influence results, as some studies have used plasma BDNF measurements, while ours focused on serum levels, which may yield different sensitivities in detecting changes [48,49]

A plausible explanation for these observations is that cognitive exercises may stimulate neuroplasticity, leading to increased BDNF secretion. This elevation in BDNF could support neuronal growth and maintenance, thereby enhancing cognitive functions and offering neuroprotective benefits. Studies have demonstrated that cognitive training, especially when combined with physical exercise, can lead to significant increases in BDNF levels, which are associated with improvements in cognitive performance. Moreover, aerobic exercises such as brisk walking and high-intensity interval training, as well as activities like yoga and dancing, have been shown to improve cognitive function and increase BDNF levels. These activities promote neuroplasticity and may alleviate symptoms associated with cognitive decline [50].

A particularly relevant finding was the increase in the number of participants who reached higher performance levels after the intervention. This improvement in functional classification suggests that training not only optimizes performance in specific tasks but may also contribute to greater autonomy in activities of daily living [11]. This finding aligns with previous studies, including that of [9], who reported that cognitive stimulation programs can translate into long-term functional benefits in older adults, especially if they already have some type of cognitive impairment.

In this sense, the findings have a clear implication in indicators of well-being such as depression, a significant difference was observed when comparing scores before and after cognitive stimulation (t = 2.45, *p* = 0.027). This finding suggests that the intervention had an effect in reducing depression-related symptoms in the population assessed. Such a result is in agreement with previous research that has shown that cognitive stimulation may play a key role in improving emotional well-being in older adults [8].

In relation to quality of life, the findings of this study reinforce the idea that cognitive training not only impacts cognitive function per se but also improves key components of subjective well-being in this population. Recent studies have shown that strengthening autonomy and social interaction contribute to a better perception of quality of life in older adults [10]. Thus, the implementation of this type of program could be integrated into public health strategies aimed at promoting active and healthy aging [51,52].

### 4.1. Perspectives for Clinical Practice

The results of this study underscore the relevance of structured cognitive stimulation programs as an effective strategy to promote cognitive health, psychological well-being, autonomy, and quality of life in older adults. The cognitive stimulation program developed and implemented in this study could represent a practical tool that can be integrated into routine geriatric nursing care.

From a clinical perspective, nursing professionals play a key role in identifying cognitive decline, initiating preventive interventions, and supporting sustained engagement in cognitive activities. The structured nature of the program allows nurses to plan and deliver sessions systematically, adapting the pace to each participant’s abilities while maintaining standardization of content [53]. This model, inspired by chronic disease management frameworks, allows for ongoing monitoring, progressive challenge, and personalized support, like approaches used in managing conditions like diabetes and hypertension, where adherence and patient-centered care are crucial [54].

Incorporating this cognitive training program into nursing practice contributes not only to cognitive and emotional improvements but also reduces the risk of functional decline and dependence, potentially lowering long-term care costs. Furthermore, by promoting active participation in cognitive, social, and recreational activities, these interventions help prevent isolation and depression among older adults [55].

Nurses are therefore positioned as central facilitators in the delivery of cognitive stimulation, ensuring that such programs are accessible, effective, and sustainable in institutional and community settings. The program presented in this study offers a structured, replicable framework that nursing professionals can use to enhance cognitive resilience, support healthy aging, and strengthen social integration in geriatric populations.

### 4.2. Limitations

While the results of this study are encouraging and provide preliminary evidence of the benefits of an individual-based cognitive stimulation program for older Mexican adults, there are some limitations that should be considered. Although participants were randomized, the small sample size and recruitment from a single gerontological center may not fully represent the broader older adult population, limiting the generalizability of the findings. Additionally, the pretest–posttest design and the lack of long-term follow-up restrict the ability to assess the durability of the observed effects. Further research with larger and more diverse samples is necessary, along with studies that explore the long-term outcomes of these interventions and evaluate their scalability across different healthcare settings to ensure their effectiveness and sustainability at a population level. Finally, the control group did not receive any type of session, which could influence the results, given the lack of control over the effect of simple participation or interaction. Consequently, the observed positive effects on autonomy and quality of life might be partially attributed to increased motivation or expectation effects rather than actual cognitive improvements. For future studies, it is important for the control group to receive structured interventions with the same frequency and duration but without cognitive training in order to better isolate the specific effect of the intervention.

In terms of cost-effectiveness, the implemented face-to-face cognitive stimulation program requires significant investment in human resources, printed materials, and physical spaces. However, it promotes social interaction and close support, which can be beneficial in populations with limited digital literacy. In contrast, computerized cognitive training offers a more adaptable and lower-cost option, with automatic difficulty adjustments, scalability, and reduced operational costs. Nevertheless, technological barriers in older adults may limit its applicability. Future studies should compare the efficacy and cost-effectiveness of both approaches, considering contextual factors that affect their real-world implementation.

## 5. Conclusions

The findings of this study suggest that the cognitive stimulation program used represents a promising strategy for improving executive functions, psychological well-being, autonomy in activities of daily living, and quality of life in older adults. The observed increase in serum BDNF levels in the intervention group provides a potential neurobiological explanation for these improvements, supporting the hypothesis that cognitive stimulation promotes neuroplasticity and brain health.

## Figures and Tables

**Figure 1 nursrep-15-00151-f001:**
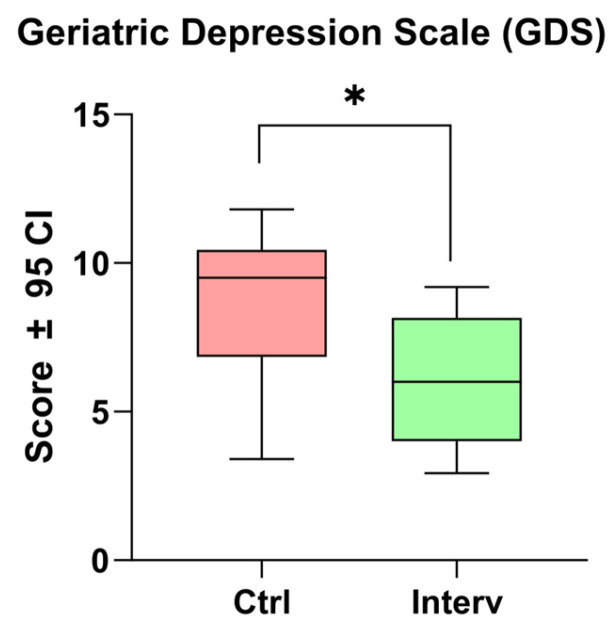
Comparison of depression scores before and after the cognitive stimulation program using the Yesavage Scale. Unpaired *t*-tests were used to measure statistical significance * *p* < 0.05.

**Figure 2 nursrep-15-00151-f002:**
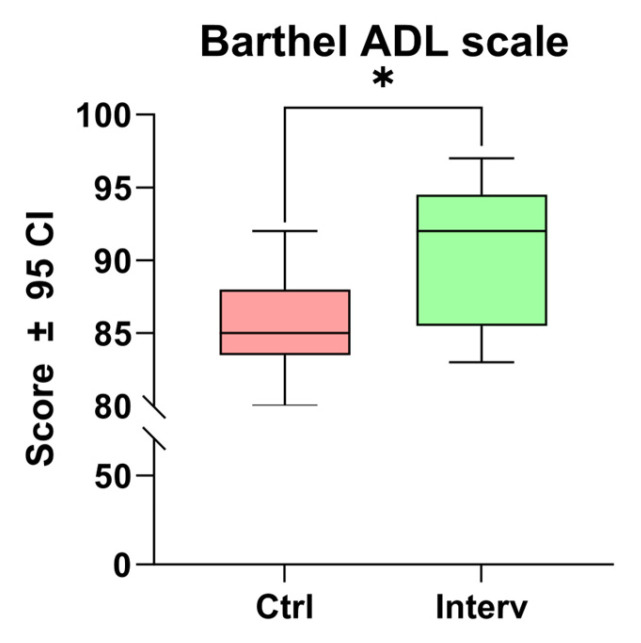
Comparison of autonomy scores in ADL before and after the cognitive stimulation program using the Barthel Index. Unpaired *t*-tests were used to measure statistical significance * *p* < 0.05.

**Figure 3 nursrep-15-00151-f003:**
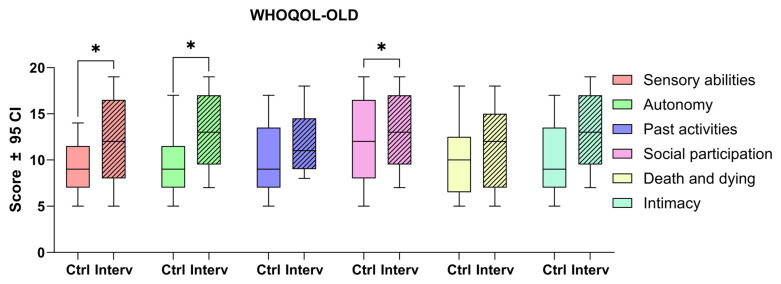
Comparison of quality of life scores between the intervention and control groups using the WHOQOL-OLD questionnaire to assess the impact of the cognitive stimulation program. Unpaired *t*-tests were used to measure statistical significance * *p* < 0.05.

**Figure 4 nursrep-15-00151-f004:**
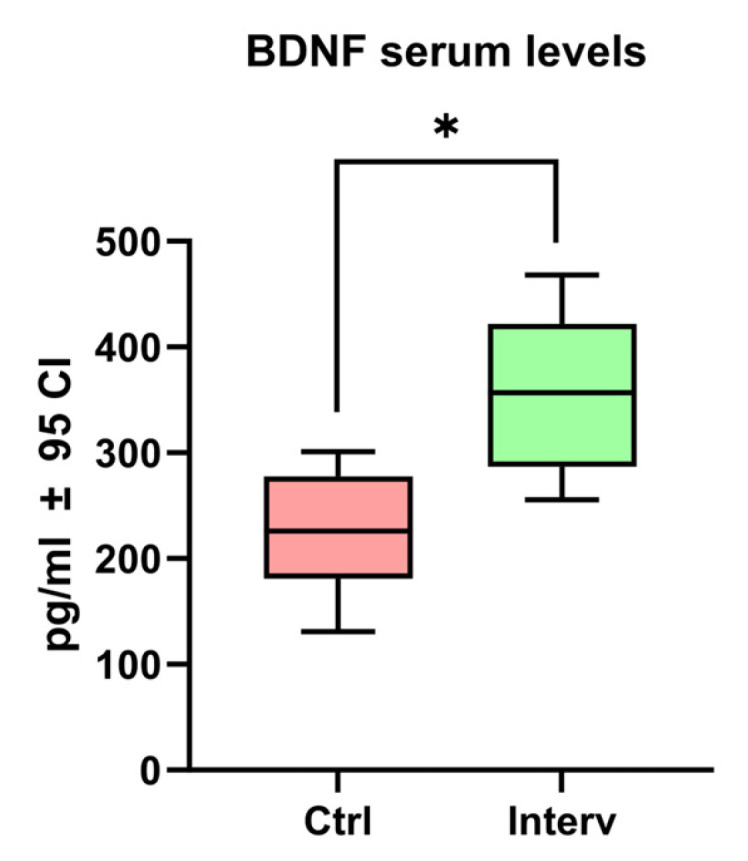
BDNF serum levels among both older adult groups. Unpaired *t*-tests were used to measure statistical significance * *p* < 0.05.

**Table 1 nursrep-15-00151-t001:** Structure of the individual cognitive stimulation program.

Content	Duration	Activities
Session introduction	5 min	Greetings/welcome. Mood check. Communicate the aim of the session.
Reality orientation	10 min	Identify time elements and address spatial elements (day of the week, month, day of the month, year, season of the year and weather conditions), using a time orientation chart.
Stimulation of cognitive domains	40 min	Explore cognitive stimulation material regarding domains as cognitive flexibility, inhibitory control, working memory, planning, reasoning, attention, concentration and processing speed.
Session closure	5 min	Analyze the difficulties, interests and benefits of the session. Goodbye.

**Table 2 nursrep-15-00151-t002:** Structure of the cognitive training manual.

Name of the Activity	Mainly Stimulated Cognitive Function	Level 1	Level 2	Level 3
Verbal Analogies	Reasoning, WM	Exercise 1.1 to 1.10	Exercise 1.11 to 1.20	Exercise 1.21 to 1.30.
2. Crossword Puzzle	WM, Attention, Reasoning	Exercise 2.1 to 2.10	Exercise 2.11 to 2.20	Exercise 2.21 to 2.30.
3. Sequence of Figures	Cognitive flexibility, Attention, Reasoning	Exercise 3.1 to 3.10	Exercise 3.11 to 3.20	Exercise 3.21 to 3.30.
4. Sequence of Numbers	Reasoning, WM, Reasoning	Exercise 4.1 to 4.10	Exercise 4.11 to 4.20	Exercise 4.21 to 4.30
5. Word Search	Attention, Processing speed, Reasoning	Exercise 5.1 to 5.10	Exercise 5.11 to 5.20	Exercise 5.21 to 5.30
6. Riddles	Reasoning, WM, Reasoning	Exercise 6.1 to 6.10	Exercise 6.11 to 6.20	Exercise 6.21 to 6.30
7. Shopping	Reasoning, Planning, Reasoning	Exercise 7.1 to 7.10	Exercise 7.11 to 7.20	Exercise 7.21 to 7.30
8. Replica	WM, Attention	Exercise 8.1 to 8.10	Exercise 8.11 to 8.20	Exercise 8.21 to 8.30
9. The Shadow	Visual perception, Attention	Exercise 9.1 to 9.10	Exercise 9.11 to 9.20	Exercise 9.21 to 9.30
10. The Mazes	Planning, Cognitive flexibility	Exercise 10.1 to 10.10	Exercise 10.11 to 10.20	Exercise 10.21 to 10.30
11. Calculating	Reasoning, Attention	Exercise 11.1 to 11.10	Exercise 11.11 to 11.20	Exercise 11.21 to 11.30
12. Which One is the Same?	Visual perception, Attention	Exercise 12.1 to 12.10	Exercise 12.11 to 12.20	Exercise 12.21 to 12.30
13. The Grid	Attention, WM	Exercise 13.1 to 13.10	Exercise 13.11 to 13.20	Exercise 13.21 to 13.30
14. Locate Yourself	Attention, Spatial reasoning	Exercise 14.1 to 14.10	Exercise 14.11 to 14.20	Exercise 14.21 to 14.30
15. Replicate the Image	WM, Visual perception	Exercise 15.1 to 15.10	Exercise 15.11 to 15.20	Exercise 15.21 to 15.30
16. Image Pairing	WM, Attention	Exercise 16.1 to 16.10	Exercise 16.11 to 16.20	Exercise 16.21 to 16.30
17. Image Selection	WM, Processing speed	Exercise 17.1 to 17.10	Exercise 17.11 to 17.20	Exercise 17.21 to 17.30
18. Words	WM, Processing speed	Exercise 18.1 to 18.10	Exercise 18.11 to 18.20	Exercise 18.21 to 18.30
19. Shapes and Colors	WM, Processing speed	Exercise 19.1 to 19.10	Exercise 19.11 to 19.20	Exercise 19.21 to 19.30
20. Giving Order	WM, Spatial reasoning	Exercise 20.1 to 20.10	Exercise 20.11 to 20.20	Exercise 20.21 to 20.30
21. Do You Remember?	WM, Attention	Exercise 21.1 to 21.10	Exercise 21.11 to 21.20	Exercise 21.21 to 21.30
22. Find the Differences	Attention, Visual perception	Exercise 22.1 to 22.10	Exercise 22.11 to 22.20	Exercise 22.21 to 22.30
23. Animated Maze	Cognitive flexibility, Planning, Reasoning	Exercise 23.1 to 23.10	Exercise 23.11 to 23.20	Exercise 23.21 to 23.30
24. Sudoku	Reasoning, WM	Exercise 24.1 to 24.10	Exercise 24.11 to 24.20	Exercise 24.21 to 24.30
25. Counting	Attention, WM	Exercise 25.1 to 25.10	Exercise 25.11 to 25.20	Exercise 25.21 to 25.30
26. The Neighbors	Attention, Processing speed, WM	Exercise 26.1 to 26.10	Exercise 26.11 to 26.20	Exercise 26.21 to 26.30
27. The Story	Reasoning, WM	Exercise 27.1 to 27.10	Exercise 27.11 to 27.20	Exercise 27.21 to 27.30
28. What Could It Be?	Attention, Visual perception	Exercise 28.1 to 28.10	Exercise 28.11 to 28.20	Exercise 28.21 to 28.30

WM: Working memory.

**Table 3 nursrep-15-00151-t003:** Board game uses in the intervention.

Stimulated Cognitive Function	Board Game
Cognitive flexibility	UNO^®^, dominoes, loteria, Chinese chopsticks, snakes and ladders, the game of the goose, marbles, bowling and take it all.
Inhibitory control	UNO^®^, dominoes, loteria, Chinese chopsticks, snakes and ladders, the game of the goose, foosball, marbles, jenga^®^ and take it all,
Working memory	UNO^®^, dominoes, lottery, Chinese chopsticks, snakes and ladders and the game of the goose
Planning	UNO^®^, dominoes, snakes and ladders and bowling.
Reasoning	UNO^®^, dominoes, loteria, Chinese chopsticks, snakes and ladders, the game of the goose, foosball, marbles, jenga, take it all and golfito.
Attention	UNO^®^, dominoes, loteria, Chinese chopsticks, snakes and ladders, the game of the goose, foosball, marbles, bowling, jenga^®^, take it all and golfito
Concentration and processing speed	UNO^®^, dominoes, loteria, Chinese chopsticks, snakes and ladders, the game of the goose, foosball, marbles, bowling, jenga^®^, take it all and golfito.

**Table 4 nursrep-15-00151-t004:** Sociodemographic characterization of the sample.

Variable		Control Group (n = 12)	Intervention Group (n = 24)	*p*
Age (years) ^a^		66.17 ± 2.93	67.75 ± 4.75	0.732 ^d^
Gender ^b^	Women	6 (50)	12 (50)	0.982 ^c^
	Men	6 (50)	12 (50)	0.973 ^c^
Level education ^c^	Primary School	2 (16.67)	4 (16.67)	0.181 ^c^
	Secondary School	2 (16.67)	4 (16.67)	0.985 ^c^
	High School	2 (16.67)	2 (8.33)	0.075 ^c^
	Technical School	0 (0)	2 (8.33)	0.053 ^c^
	High School	4 (33.33)	10 (41.67)	0.184 ^c^
	Postgraduate	2 (16.67)	2 (8.33)	0.089 ^c^
Initial cognitive assessment score ^d^	Total executive functions	73.83 ± 20.63	80.75 ± 44.86	0.461 ^d^
	Orbitofrontal cortex	75.50 ± 25.22	73.82 ± 24.77	0.775 ^d^
	Anterior prefrontal cortex	83.67 ± 28.97	82.64 ± 25.33	0.311 ^d^
	Dorsolateral cortex	82.17 ± 15.48	90.18 ± 14.84	0.285 ^d^

^a^ Data presented as mean ± standard deviation. ^b^ Data presented as frequencies and percentages. ^c^ Comparison using the X^2^ test. ^d^ Comparison using the *t*-test.

**Table 5 nursrep-15-00151-t005:** Intragroup comparison between cognitive assessments.

BANFE-3		Control Group (n = 12)			Intervention Group (n = 24)		
Zone	Classification	Pre Evaluation	Post Evaluation	*p*	Pre Evaluation	Post Evaluation	*p*
Orbitofrontal	Score ^a^	75.50 ± 25.22	71.50 ± 14.77	0.832 ^d^	73.82 ± 24.77	85.17 ± 26.75	0.002 ^d^
	High normal ^b^	0 (0)	0 (0)	-	0 (0)	0 (0)	-
	Normal ^b^	6 (50)	6 (50)	0.932 ^c^	2 (8.34)	12 (50)	0.009 ^c^
	Mild–Mod impairment ^b^	2 (16.67)	2 (16.67)	0.948 ^c^	6 (25)	10 (41.66)	0.921 ^c^
	Severe impairment ^b^	4 (33.33)	4 (33.33)	0.931 ^c^	16 (66.66)	2 (8.34)	0.012 ^c^
Anterior prefrontal	Score ^a^	83.67 ± 28.97	82.64 ± 25.33	0.124 ^d^	81.14 ± 26.22	100.28 ± 28.13	0.001 ^d^
	High normal ^b^	2 (16.67)	2 (16.67)	0.913 ^c^	2 (8.33)	8 (33.33)	0.018 ^c^
	Normal ^b^	2 (16.67)	2 (16.67)	0.974 ^c^	14 (58.33)	10 (41.67)	0.045 ^c^
	Mild–Mod impairment ^b^	4 (33.33)	6 (50)	0.113 ^c^	2 (8.33)	4 (16.67)	0.948 ^c^
	Severe impairment ^b^	4 (33.33)	2 (16.67)	0.081 ^c^	6 (25)	2 (8.33)	0.214 ^c^
Dorsolateral	Score ^a^	82.17 ± 15.48	90.18 ± 14.84	0.372 ^d^	80.28 ± 18.54	112.67 ± 28.17	0.031 ^d^
	High normal ^b^	2 (16.67)	2 (16.67)	0.948 ^c^	2 (8.33)	2 (8.33)	0.965 ^c^
	Normal ^b^	4 (66.67)	6 (50)	0.642 ^c^	16 (66.67)	16 (66.67)	0.924 ^c^
	Mild–Mod impairment ^b^	2 (16.67)	2 (16.67)	0.985 ^c^	2 (8.33)	6 (25)	0.017 ^c^
	Severe impairment ^b^	4 (33.33)	2 (16.67)	0.271	4 (16.67)	0 (0)	0.003 ^c^
Total executive functions	Score ^a^	73.83 ± 20.63	76.83 ± 11.89	0.823 ^d^	80.75 ± 44.86	93.17 ± 19.66	0.001 ^d^
	High normal ^b^	0 (0)	0 (0)	-	2 (8.33)	2 (8.33)	0.924 ^c^
	Normal ^b^	6 (50)	6 (50)	0.924 ^c^	6 (25)	12 (50)	0.001 ^c^
	Mild–Mod impairment ^b^	2 (16.67)	4 (33.33)	0.073 ^c^	12 (50)	6 (33.34)	0.438 ^c^
	Severe impairment ^b^	4 (33.33)	2 (16.67)	0.091 ^c^	4 (16.67)	2 (8.33)	0.137 ^c^

^a^ Data presented as mean ± standard deviation. ^b^ Data presented as frequencies and percentages. ^c^ Comparison using the Wilcoxon test. ^d^ Comparison using the repeated-measures *t*-test.

**Table 6 nursrep-15-00151-t006:** Intergroup comparison between post-cognitive assessment.

BANFE-3		Post Evaluation		*p*
Zone	Classification	Control Group (n = 12)	Intervention Group (n = 24)	
Orbitofrontal	Score ^a^	71.50 ± 14.77	85.17 ± 26.75	0.001 ^d^
	High normal ^b^	0 (0)	0 (0)	-
	Normal ^b^	6 (50)	12 (50)	0.023 ^c^
	Mild–Mod impairment ^b^	2 (16.67)	10 (41.66)	0.014 ^c^
	Severe impairment ^b^	4 (33.33)	2 (8.34)	0.128 ^c^
Anterior prefrontal	Score ^a^	82.64 ± 25.33	100.28 ± 28.13	0.031 ^d^
	High normal ^b^	2 (16.67)	8 (33.33)	0.041 ^c^
	Normal ^b^	2 (16.67)	10 (41.67)	0.017 ^c^
	Mild–Mod impairment ^b^	6 (50)	4 (16.67)	0.016 ^c^
	Severe impairment ^b^	2 (16.67)	2 (8.33)	0.928 ^c^
Dorsolateral	Score ^a^	90.18 ± 14.84	112.67 ± 28.17	0.009 ^d^
	High normal ^b^	2 (16.67)	2 (8.33)	0.352 ^c^
	Normal ^b^	6 (50)	16 (66.67)	0.184 ^c^
	Mild–Mod impairment ^b^	2 (16.67)	6 (25)	0.017 ^c^
	Severe impairment ^b^	2 (16.67)	0 (0)	0.008 ^c^
Total executive functions	Score ^a^	76.83 ± 11.89	93.17 ± 19.66	0.028 ^d^
	High normal ^b^	0 (0)	2 (8.33)	0.031 ^c^
	Normal ^b^	6 (50)	12 (50)	0.004 ^c^
	Mild–Mod impairment ^b^	4 (33.33)	8 (33.34)	0.040 ^c^
	Severe impairment ^b^	2 (16.67)	2 (8.33)	0.005 ^c^

^a^ Data presented as mean ± standard deviation. ^b^ Data presented as frequencies and percentages. ^c^ Comparison using the Wilcoxon test. ^d^ Comparison using the repeated-measures *t*-test.

## Data Availability

Data is contained within the article or Appendix A.

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
