# Peer review of "Cognitive Stimulation and Its Effects on Well-Being, Executive Functions, and Brain-Derived Neurotrophic Factor in Older Adults from a Mexican Geriatric Center: A Quasi-Experimental Study"

_nursrep, 2025, doi:10.3390/nursrep15050151_

Round 1
Reviewer 1 Report
Comments and Suggestions for Authors
The article entitled "Impact of cognitive training in older adults: well-being, executive functions and BDNF levels" explores the impact of cognitive training in older adults, evaluating its effects on psychological well-being, executive function performance and BDNF levels. This study presents a structure and content suitable for publication, but the following aspects must first be addressed:
- In the text in general, sentences with confusing constructions or grammatical errors can be observed that make reading difficult. Please review.
- In the introduction, the importance of cognitive stimulation is discussed, but it would be necessary to first present a clear cohesion from the problem of aging.
- Although the instruments used to measure the variables are described, it is necessary to explain their applicability in the population studied.
- The procedure for assigning participants to groups is not clearly specified. Although it is mentioned that they were matched on variables such as age and education, the exact method should be detailed (was it random?
- In the description of the cognitive stimulation program, it would be useful to explain how it was adapted to different levels of cognitive decline and whether there were adjustments based on the progress of the participants.
- The use of t-tests and Wilcoxon is mentioned, but it is not specified whether any correction for multiple comparisons was made, which is important when analyzing multiple variables.
- Although previous studies are mentioned, a deeper analysis of how the results of the study compare to similar research is lacking. For example, it would be useful to discuss to what extent the findings on BDNF match previous studies and whether there are differences attributable to the methodology used.
- While some limitations are acknowledged (such as sample size), others could be included, such as possible selection bias (since participants were recruited at a single center).
- Some statements in the conclusion could be moderated. It is mentioned that the program "could be an effective strategy," which is valid, but the need to replicate these findings in studies with larger samples and long-term follow-up should be emphasized.
Comments on the Quality of English Language
It is recommended to review the English of the article, as some expressions are not completely understandable and contain grammatical or spelling errors. This may affect the clarity of the message and the fluidity of the text.
Author Response
We sincerely thank the reviewers for their valuable comments and suggestions, which have significantly contributed to improving the quality and clarity of our manuscript. Below we respond to each of their comments:
In the text in general, sentences with confusing constructions or grammatical errors can be observed that make reading difficult. Please review.
Answer: We have carefully revised the English style and improved the overall clarity of the document. Additionally, we have thoroughly reviewed it to ensure the text is clear and free of grammatical errors. Thank you for your valuable feedback
In the introduction, the importance of cognitive stimulation is discussed, but it would be necessary to first present a clear cohesion from the problem of aging.
Answer: We appreciate the comment. We have modified the introduction to include the relationship between aging and cognitive stimulation in the context of our study.
Although the instruments used to measure the variables are described, it is necessary to explain their applicability in the population studied.
Answer: Thanks for the comment. We've added the rationale for using each instrument, including Cronbach's alpha.
The procedure for assigning participants to groups is not clearly specified. Although it is mentioned that they were matched on variables such as age and education, the exact method should be detailed (was it random?.
Answer: Thank you for your valuable observation. To ensure comparability between groups, we employed a stratified randomization procedure. Participants were stratified based on sex, age, educational level, and pre-intervention cognitive function score. We have clarified this aspect in the revised manuscript.
In the description of the cognitive stimulation program, it would be useful to explain how it was adapted to different levels of cognitive decline and whether there were adjustments based on the progress of the participants.
Answer: The cognitive stimulation program was not specifically adapted to different levels of cognitive decline; instead, it was structured around three predefined levels of complexity (basic, intermediate, and advanced), designed to progressively challenge cognitive functions. Additionally, there were no adjustments made based on the individual progress of participants throughout the intervention. However, the program allowed flexibility within each session: if a participant completed an activity before, they could immediately proceed to the next task without having to wait for the rest of the group. This approach aimed to maintain engagement and accommodate individual pacing within the structured framework.
The use of t-tests and Wilcoxon is mentioned, but it is not specified whether any correction for multiple comparisons was made, which is important when analyzing multiple variables.
Answer: Thank you for your insightful comment. To address the concern regarding multiple comparisons and minimize the risk of Type I error, we applied the Bonferroni correction where appropriate. We have clarified this methodological detail in the revised manuscript.
Although previous studies are mentioned, a deeper analysis of how the results of the study compare to similar research is lacking. For example, it would be useful to discuss to what extent the findings on BDNF match previous studies and whether there are differences attributable to the methodology used.
Answer: Thank you for your valuable feedback. We have incorporated a more in-depth analysis comparing our findings to previous research, particularly focusing on the results related to BDNF. We believe this addition enhances the context and significance of our results.
While some limitations are acknowledged (such as sample size), others could be included, such as possible selection bias (since participants were recruited at a single center).
Answer: Thank you for your comments. We've expanded the study's limitations (lines 376 to 380).
Some statements in the conclusion could be moderated. It is mentioned that the program "could be an effective strategy," which is valid, but the need to replicate these findings in studies with larger samples and long-term follow-up should be emphasized.
Answer: We appreciate the suggestion. We have modified the conclusion and we've expanded the study's limitations (lines 376 to 380).
Reviewer 2 Report
Comments and Suggestions for Authors
This quasi-experimental study investigates the impact of a six-month intervention involving near-daily cognitive training on executive functioning, well-being, and BDNF expression in older adults. While the research report is well-structured, the manuscript requires some major revisions to address the concerns identified. Additionally, minor adjustments will be convenient prior to its publication.
Major
- Revise the structure of the introduction, since several of the outcome measures refer to well-being, yet it is not mentioned in the introduction. I suggest providing more background on this aspect.
- The control group did not attend any type of session. This design is flawed because it does not guarantee that the differences are not due to the mere fact of having attended the sessions. For this, the control group should have attended the same number of times and should have carried out activities of the same duration but not involving cognitive training. On the other hand, it is unclearly mentioned that the control group does not attend the usual medical care. Therefore, the comparison between the groups may not be due to the intervention (cognitive training) but to the differences between elderly people who do or do not regularly attend the care center.
- In the results related to Well-being indicators, the text mentions pre-test vs. post-test differences for the intervention group, which does not seem entirely consistent with the graphs in which the comparison between intervention and control groups is represented. A clarification or better explanation is required to show consistency between the text and the graphs.
- More bibliographic references are required to support the variations in BDNF expression as a consequence of plastic changes due to cognitive training. As well as specific references of these variations in older adult population.
Minor
- The abstract should be rephrased better. It starts directly with the objective of the study without introducing at least one sentence of the background.
- It needs revision of the English grammar, for example the paragraph that includes lines 52-54.
- In line 152 it is necessary to detail more precisely the validity and reliability of the battery. In addition, a bibliographic reference is needed.
- Improve the quality of figures 1, 2, and 4 because the texts overlap and the legend of the graphs is not visible.
- Figure 3 is missing.
- More detail is needed on the limitations of the study, which are not only due to the sample size but also to the research design.
Author Response
We sincerely thank the reviewers for their valuable comments and suggestions, which have significantly contributed to improving the quality and clarity of our manuscript. Below we respond to each of their comments:
Major
Revise the structure of the introduction, since several of the outcome measures refer to well-being, yet it is not mentioned in the introduction. I suggest providing more background on this aspect.
Answer: Thank you for your feedback. We've modified the introduction, highlighting the relevance of the "well-being" variables used in the study.
The control group did not attend any type of session. This design is flawed because it does not guarantee that the differences are not due to the mere fact of having attended the sessions. For this, the control group should have attended the same number of times and should have carried out activities of the same duration but not involving cognitive training. On the other hand, it is unclearly mentioned that the control group does not attend the usual medical care. Therefore, the comparison between the groups may not be due to the intervention (cognitive training) but to the differences between elderly people who do or do not regularly attend the care center.
Answer: We appreciate your comment. First, in line 93 we clarified that both groups continued to receive their usual medical and therapeutic care during the study. Second, we acknowledged the fact that the control group did not receive any sessions (lines 381 to 383) as a limitation. However, one positive aspect of the design used is that it allowed us to observe the impact of cognitive training under real-life conditions, comparing it with a group that did not receive any additional intervention. This reinforces the applicability of the findings in clinical and community settings where more complex controls are not always possible. Furthermore, the magnitude of the observed effects suggests that the intervention had a relevant impact beyond the simple effect of participation.
In the results related to Well-being indicators, the text mentions pre-test vs. post-test differences for the intervention group, which does not seem entirely consistent with the graphs in which the comparison between intervention and control groups is represented. A clarification or better explanation is required to show consistency between the text and the graphs.
Answer: Thank you for your observation. We have revised the text to ensure consistency with the graphs, explicitly mentioning the comparison between the intervention and control groups. This adjustment provides a clearer and more accurate representation of the results.
More bibliographic references are required to support the variations in BDNF expression as a consequence of plastic changes due to cognitive training. As well as specific references of these variations in older adult population.
Answer: Thank you for your valuable comment. We have included additional bibliographic references to provide a more comprehensive comparison and explanation regarding the changes in BDNF expression because of cognitive training. Specifically, we have also included studies that focus on variations in BDNF in the older adult population, as requested.
Minor
The abstract should be rephrased better. It starts directly with the objective of the study without introducing at least one sentence of the background.
Answer: Thanks for the feedback. We've modified the abstract.
It needs revision of the English grammar, for example the paragraph that includes lines 52-54.
Answer: We have revised the text to improve the grammar and clarity.
In line 152 it is necessary to detail more precisely the validity and reliability of the battery. In addition, a bibliographic reference is needed.
Answer: Thanks for the comment. We've added the rationale for using each instrument, including Cronbach's alpha.
Improve the quality of figures 1, 2, and 4 because the texts overlap and the legend of the graphs is not visible.
Answer: Thank you for your valuable feedback. The figures are correctly formatted in the Word document, but it seems that the automatic conversion to PDF may have caused the text overlap and the legend to become unclear.
Figure 3 is missing.
Answer: Thank you for your observation. Figure 3 is included in the Word document, but it appears that the automatic formatting in the PDF version may have caused its omission. The figure is correctly presented in the original manuscript.
More detail is needed on the limitations of the study, which are not only due to the sample size but also to the research design.
Answer: Thank you for your comments. We've expanded the study's limitations (lines 376 to 380).
Reviewer 3 Report
Comments and Suggestions for Authors
The manuscript presents a study with significant medical and social implications. While there is extensive research on cognitive stimulation in older adults, the inclusion of biomarkers such as BDNF as complementary measures to support behavioral assessments is relatively uncommon, as is the consideration of emotional aspects. The article is well-structured, with a clear, direct, and well-supported narrative. I believe it is suitable for publication following the incorporation of the following revisions.
- The assessment procedure should be specified in greater detail. Were the three well-being metrics administered in the same session as the cognitive assessment? Were they conducted by a neuropsychologist, or were they self-administered? If administered simultaneously, lines 146–147 may fit better under section 2.1. Also, clarify how the blood analysis was conducted?
- Lines 165–166 appear redundant, as "to assess depression" is stated twice.
- Further clarification is needed regarding the structure and dynamics of the intervention. Was there a predefined pattern for task assignment per session? Was the task order within each session standardized or individualized? Did participants complete the same number of activities per session, regardless of their pace of completion? How was the difficulty level applied—was it progressively increasing or dynamically adjusted based on performance? If dynamic, was there a possibility of repeating previously completed activities? The supplementary material suggests a fixed structure for all participants, but it is unclear whether this was the actual implementation or just an illustrative example.
- There appears to be an inconsistency between the text description and the figure labels. The text states that the analysis compares results "before and after the cognitive stimulation program," suggesting a within-group pre-post comparison. However, the figures are labeled as “Ctrl” and “Interv”, implying a between-group comparison. Please clarify whether the analysis is within-group, between-group, or both, and adjust either the figure labels or the text for consistency.
- Is Figure 3 missing?
- Effect size calculations are mentioned in the methodology, but they are not consistently reported alongside p-values and pre-post differences in the results. Additionally, while the discussion highlights statistically significant improvements, it does not examine the strength of these effects or their clinical relevance. Distinguishing between statistical significance and the actual magnitude of the changes observed would provide a more meaningful interpretation of the findings.
- The discussion lacks a practical comparison of efficacy and efficiency. While the intervention is comprehensive and appropriate, it also involves high costs. How does this intervention compare with computerized cognitive training, which may offer greater adaptability and lower costs? A brief discussion on the cost-benefit aspect and its implications for real-world implementation would enhance the study’s applicability.
- Given that the control group was inactive, alternative explanations for the effects observed should be considered. For example, could the positive effects on autonomy and quality of life be driven more by increased motivation or expectancy effects rather than actual cognitive improvements?
Author Response
We sincerely thank the reviewers for their valuable comments and suggestions, which have significantly contributed to improving the quality and clarity of our manuscript. Below we respond to each of their comments:
The assessment procedure should be specified in greater detail. Were the three well-being metrics administered in the same session as the cognitive assessment? Were they conducted by a neuropsychologist, or were they self-administered? If administered simultaneously, lines 146–147 may fit better under section 2.1. Also, clarify how the blood analysis was conducted?
Answer: First, the blood sample was collected (lines 128–129). Afterward, participants were allowed to have breakfast and consume their regular medication. Subsequently, after a minimum period of one hour had elapsed, the cognitive assessment was conducted (lines 137–152). On the following day, well-being indicators were evaluated. Both the cognitive assessment and the evaluation of well-being indicators were carried out by two neuropsychologists who were blinded to the study objectives and to the condition assigned to each participant.
Lines 165–166 appear redundant, as "to assess depression" is stated twice.
Answer: We appreciate the feedback. We've removed the duplicate information.
Further clarification is needed regarding the structure and dynamics of the intervention. Was there a predefined pattern for task assignment per session? Was the task order within each session standardized or individualized? Did participants complete the same number of activities per session, regardless of their pace of completion? How was the difficulty level applied—was it progressively increasing or dynamically adjusted based on performance? If dynamic, was there a possibility of repeating previously completed activities? The supplementary material suggests a fixed structure for all participants, but it is unclear whether this was the actual implementation or just an illustrative example.
Answer: We appreciate your valuable comment. Below, we provide greater clarity regarding the structure and dynamics of the intervention: The intervention was designed with a predefined and standardized pattern for task assignment in each session. Each session followed a fixed daily schedule for all participants (see Table 1 and Appendix A.2), which specified the order and type of activities to be completed (board games and paper-based activities), as well as the time allocated to each component. Therefore, the order of tasks within each session was standardized and not individualized. All participants completed the same number of activities per session, regardless of their individual pace. However, they were allowed to progress through the activities at their own rhythm during each session, with the support of a nursing student available to provide guidance if needed. Regarding the level of difficulty, it was not dynamically adjusted based on performance but was implemented in a progressive and standardized manner. The cognitive training manual included 28 types of activities, each with three predefined levels of complexity (basic, intermediate, and advanced). Progression through these levels followed a sequential and programmed order: participants were required to complete the basic level exercises first, followed by the intermediate level, and finally the advanced level (see Table 2). Previously completed activities were not repeated, as the structure was designed for progressive advancement without regressions or dynamic adjustments. Finally, Appendix A.1 (Example of a Level 1 exercise for each of the 28 activities) accurately represents the 28 activities that were designed; however, for illustrative purposes, only one level 1 exercise corresponding to each of these 28 activities is presented as an example.
There appears to be an inconsistency between the text description and the figure labels. The text states that the analysis compares results "before and after the cognitive stimulation program," suggesting a within-group pre-post comparison. However, the figures are labeled as “Ctrl” and “Interv”, implying a between-group comparison. Please clarify whether the analysis is within-group, between-group, or both, and adjust either the figure labels or the text for consistency.
Answer: Thank you for bringing this inconsistency to our attention. Upon review, we identified that the correct analysis was a between-group comparison, focusing on control versus intervention groups. To address this, we have updated the figure labels to accurately represent the between-group comparison and ensured that the text description aligns with this analysis. These adjustments should now provide clarity and consistency throughout the manuscript.
Is Figure 3 missing?
Answer: Answer: Thank you for your observation. Figure 3 is included in the Word document, but it appears that the automatic formatting in the PDF version may have caused its omission. The figure is correctly presented in the original manuscript.
Effect size calculations are mentioned in the methodology, but they are not consistently reported alongside p-values and pre-post differences in the results. Additionally, while the discussion highlights statistically significant improvements, it does not examine the strength of these effects or their clinical relevance. Distinguishing between statistical significance and the actual magnitude of the changes observed would provide a more meaningful interpretation of the findings.
Answer: Thank you for highlighting these important considerations. In response, we have thoroughly reviewed the results and discussion sections to ensure consistency in reporting effect size calculations alongside p-values and pre-post differences. Additionally, we have revised the discussion to include an in-depth examination of the strength of the observed effects and their clinical relevance. By distinguishing between statistical significance and the magnitude of the changes, we aim to provide a more meaningful interpretation of our findings, offering a clearer understanding of their practical implications.
The discussion lacks a practical comparison of efficacy and efficiency. While the intervention is comprehensive and appropriate, it also involves high costs. How does this intervention compare with computerized cognitive training, which may offer greater adaptability and lower costs? A brief discussion on the cost-benefit aspect and its implications for real-world implementation would enhance the study’s applicability.
Answer: In response to your suggestion, we have incorporated a paragraph in the discussion section comparing the face-to-face cognitive stimulation program used with computerized cognitive training alternatives. In this section, we highlight that, although the in-person intervention fosters social interaction and close support — aspects that are particularly valuable in populations with limited digital literacy — it involves higher costs and limited scalability. In contrast, computerized cognitive training programs offer greater adaptability, lower operational costs, and broader scalability, although technological barriers may limit their effectiveness in certain contexts. Finally, we emphasize the need for future studies to directly compare the efficacy and cost-effectiveness of both approaches, taking into account contextual factors that may influence their real-world implementation.
Given that the control group was inactive, alternative explanations for the effects observed should be considered. For example, could the positive effects on autonomy and quality of life be driven more by increased motivation or expectancy effects rather than actual cognitive improvements?
Answer: We appreciate your comment and agree on the importance of considering alternative explanations for the observed effects. In response to your observation, we have incorporated this consideration into the limitations section of the manuscript, noting that the control group did not receive any type of session, which could have influenced the results due to the lack of control over possible effects derived from participation or social interaction. The following limitation was added to the text: “Finally, the control group did not receive any type of session, which could influence the results, given the lack of control over the effect of simple participation or interaction. Consequently, the observed positive effects on autonomy and quality of life might be partially attributed to increased motivation or expectation effects rather than actual cognitive improvements. For future studies, it is important for the control group to receive structured interventions with the same frequency and duration, but without cognitive training, in order to better isolate the specific effect of the intervention.”. This addition acknowledges the possibility that the positive effects observed on autonomy and quality of life might be, at least in part, the result of increased motivation or expectation effects. Therefore, we suggest that future studies implement active control groups to more precisely isolate the specific effect of the cognitive intervention.
Reviewer 4 Report
Comments and Suggestions for Authors
Dear Authors,
the comments in the annex file.
Best

Author Response
We sincerely thank the reviewers for their valuable comments and suggestions, which have significantly contributed to improving the quality and clarity of our manuscript. Below we respond to each of their comments:
Editing: The bibliography does not follow the journal's template.
Answer: Thank you for your comment. We have revised the bibliography to ensure it fully complies with the journal's template and formatting guidelines.
Title: It lacks fundamental elements such as the setting and the type of study conducted.
Answer: We appreciate the suggestion. We've modified the title to include the context and type of study conducted.
Abstract: It could benefit from more detail regarding the possible practical developments of the collected data (see also discussions), particularly in terms of nursing clinical practice.
Answer: We appreciate your comment and consider your suggestion highly relevant. In response, we have enriched the abstract by including a section that highlights the practical implications of the findings for nursing clinical practice. Specifically, we emphasize that the results reinforce the key role of nursing professionals in implementing structured cognitive stimulation programs as part of comprehensive care for institutionalized older adults. Furthermore, it is noted that these interventions can be promoted and facilitated by nurses to enhance autonomy, emotional stability, and cognitive health in this population. This addition aims to underline the practical value of the collected data and its direct applicability in gerontological care settings.
Keywords: In my opinion, there are too many and often redundant (see the first two). I suggest selecting 4 or 5 highly attractive keywords for the reader, which, like in the title, contain key elements such as the type of study conducted and the reference setting.
Answer: Thank you for your thoughtful suggestion regarding the keywords. In response, we have carefully selected 4 to 5 highly relevant keywords that encapsulate the core elements of the study, including the population studied, the intervention applied, and the key outcomes observed. These revised keywords are designed to be both concise and highly attractive to readers, reflecting the type of study conducted and the reference setting.
Introduction: Overall, it is essential but fairly well presented. The objectives are unclear. I suggest reformulating them in the classic format: "The primary aim of the study was..." and "While the secondary aims were...". Additionally, considering the importance of the conducted study, I suggest creating a dedicated section either at the end of the introduction ("1.1 Aims") or as the first element in the methods ("Aims"), adding the specific research questions.
Answer: We appreciate the comments. We have modified the wording of the study objectives.
Methods: This is the aspect that certainly deserves the most attention. The lack of a structured reporting method, such as the STROBE checklist (doi:10.1016/j.jclinepi.2007.11.008), which is mandatory for the relevant scientific community, would have also supported the authors in making the presentation of the work more fluid and structured, especially in methodological terms, even though overall it is fairly structured. Including this checklist as a supplementary file and referencing it in the text would be crucial to improving the quality and reproducibility of the study, making the manuscript more transparent and scientifically valid, especially considering it is mandatory for the relevant scientific community.
Answer: We sincerely appreciate your observation regarding the use of the STROBE checklist. In response to your valuable suggestion, we have thoroughly reviewed the manuscript to ensure that all relevant methodological aspects align with the STROBE guidelines. Additionally, we have explicitly stated in the Methods section that the study reporting was conducted in accordance with the STROBE recommendations. Furthermore, we have prepared and included the complete STROBE checklist as a supplementary file, clearly referencing it in the manuscript.
Results: This is certainly the least controversial part of the study, and I wouldn’t suggest major changes except for proposing more accessible tables (especially for table 2) that facilitate a better understanding of the conducted study. It would certainly benefit from the improvements suggested earlier and later.
Answer: We appreciate your suggestion regarding the improvement of the tables, particularly Table 2, to make the presentation of the intervention clearer and more accessible. In response, we carefully explored alternative structures and formats. However, after multiple attempts to reorganize the information, we found that simplified or alternative table structures did not allow us to present the full level of detail required to clearly convey the complexity, progression, and distribution of the cognitive training activities. Given the importance of providing precise information about the intervention structure, we concluded that the current table format is the most appropriate to ensure clarity and transparency for readers. Nonetheless, we have carefully reviewed the tables to enhance their readability and coherence, and have made minor adjustments where possible.
Discussion: Overall, decent, but it could improve from an evidence-based perspective, and (as with the Abstract) it loses some effectiveness in terms of clinical practice. In this regard, I suggest developing a specific section dedicated to, for example, "Perspectives for Clinical Practice," which could certainly make the discussion more engaging. To this end, I suggest that the authors expand the discussion on the clinical management of these patients from the perspective of nursing improvement, starting from the experience of managing chronic diseases that are very similar to the studied population, with a view to overall lifestyle improvement. For this purpose, I would like to suggest a very relevant paper: doi.org/10.3390/diabetology5040029.
Answer: We thank you for your valuable suggestion. In response, we have added a new subsection titled "Perspectives for Clinical Practice" in the Discussion. This section highlights the key role of nurses in integrating structured cognitive stimulation programs into routine care for older adults, emphasizing their contribution to cognitive health, autonomy, and overall well-being.
Limitations: I suggest creating a specific section and including the generalizability of the obtained results, not only in terms of the small sample size but also in terms of the single study setting.
Answer: We appreciate your suggestion. We've added a dedicated section to address the limitations.
Conclusions: To be restructured according to the previous suggestions.
Answer: We appreciate the comment. We've restructured the conclusions section.
Bibliography: Should be expanded according to the provided suggestions and possibly updated, especially for those older than 15/20 years unless they provide extremely strong evidence. In summary, the manuscript presents extremely interesting scientific results. According to the suggestion provided, in my humble opinion, it could have significant relevance in the relevant scientific community.
Answer: Thank you for your insightful suggestion. We have updated the references to include more recent studies, while eliminating outdated and redundant sources. We believe these changes enhance the manuscript’s relevance and alignment with current scientific literature.
Round 2
Reviewer 3 Report
Comments and Suggestions for Authors
I appreciate the authors’ thoughtful and thorough response to the comments provided in my initial review. The revised manuscript shows clear improvements in clarity, methodological transparency, and alignment between the study’s objectives and its design.
All of my previous concerns have been sufficiently addressed, and the manuscript has been strengthened as a result. Together with the revisions made in response to the other reviewers’ comments, the current version meets the standards of academic rigor, and I considered it suitable for publication.
Author Response
We sincerely thank the reviewer for their careful reading of the manuscript and for their constructive comments and suggestions.
Reviewer 4 Report
Comments and Suggestions for Authors
The comments in the annex file

Author Response
We sincerely thank the reviewer for their careful reading of the manuscript and for their constructive comments and suggestions. Each of the points raised has been carefully considered and addressed in the revised version of the manuscript. Below, we provide a detailed explanation of how each comment was attended to, along with the corresponding modifications made to the text. We believe these improvements have strengthened the quality and clarity of our work.
In the title and keywords, an acronym is used, which is not recommended for easier reading; moreover, the same acronym is used in the abstract without being explained.
Answer: Thank you for your valuable observation. In response to your comment: We have replaced the acronym "BDNF" with the full term "Brain Derived Neurotrophic Factor" in both the title and keywords, in order to improve readability and comply with the recommendation to avoid acronyms. Additionally, the use of the full term aligns with the standardized terminology established by the Health Sciences Descriptors (DeCS/MeSH), ensuring consistency with international indexing systems.
The bibliography is disorganized and poorly edited, which does not allow for a proper interpretation of the supporting references.
Answer: We sincerely appreciate your valuable feedback. In response to your suggestion, we have carefully revised and reformatted the bibliography in accordance with the ACS Style Guide, as outlined in the Instructions for Authors of Nursing Reports. We hope this improves clarity and facilitates a more accurate interpretation of the supporting references
Furthermore, it is unclear why two Strobe citations are included, one of which is placed at the end of the text.
Answer: Thank you for your observation. We have reviewed the manuscript and confirm that the STROBE citation has now been unified. We appreciate your attention to detail, which has helped us improve the clarity and coherence of the manuscript.
Ethical aspects are always mentioned in the dedicated section but not in the text.
Answer: Thank you for your valuable observation. In response, we would like to clarify that we consider it necessary to declare the ethical aspects both in the Methodology section and in the final section of the manuscript entitled "Institutional Review Board Statement", following common scientific practice and to ensure full transparency regarding the ethical approval process.
It is recommended to use "gender" more appropriately instead of "sex."
Answer: Thank you for your observation. We have reviewed the manuscript to ensure appropriate use of terminology. The term "gender" has now been used instead of "sex" in all sections, as the data were based on self-reported identity rather than biological classification.
The "Perspective for Clinical Practice" section, rightly added after my suggestion, lacks bibliographic support and is not placed in the recommended section, which should be in the discussion with numbering 4.1.
Answer: Thank you for your valuable suggestion. In response, we have added the corresponding bibliographic references to support the content in the section "Perspectives for Clinical Practice." Additionally, we have relocated this section within the Discussion, as recommended, and adjusted the section numbering to 4.1 in accordance with the journal’s formatting guidelines.
The limitations should also be considered a subsection of the discussion, and I suggest following the above order and assigning them the numbering 4.2.
Answer: We have taken the suggestion into account.
Round 3
Reviewer 4 Report
Comments and Suggestions for Authors
Dear Authors,
very good job. In this form ready for publication.
Best